# UCLID-Net: Single View Reconstruction in Object Space

**Benoit Guillard**          **Edoardo Remelli**          **Pascal Fua**
CVLab
EPFL, Switzerland
`{firstname.lastname}@epfl.ch`

## Abstract

Most state-of-the-art deep geometric learning single-view reconstruction approaches rely on encoder-decoder architectures that output either shape parametrizations [7, 8, 23] or implicit representations [14, 26, 4]. However, these representations rarely preserve the Euclidean structure of the 3D space objects exist in. In this paper, we show that building a geometry preserving 3-dimensional latent space helps the network concurrently learn global shape regularities and local reasoning in the object coordinate space and, as a result, boosts performance.

We demonstrate both on ShapeNet synthetic images, which are often used for benchmarking purposes, and on real-world images that our approach outperforms state-of-the-art ones. Furthermore, the single-view pipeline naturally extends to multi-view reconstruction, which we also show.

## 1   Introduction

Most state-of-the-art deep geometric learning Single-View Reconstruction approaches (SVR) rely on encoder-decoder architectures that output either explicit shape parametrizations [7, 8, 23] or implicit representations [14, 26, 4]. However, the representations they learn rarely preserve the Euclidean structure of the 3D space objects exist in, and rather rely on a global vector embedding of the input image at a semantic level. In this paper, we show that building a geometry preserving 3-dimensional representation helps the network concurrently learn global shape regularities and local reasoning in the object coordinate space and, as a result, boosts performance. This corroborates the observation that choosing the right coordinate frame for the output of a deep network matters a great deal [21].

In our work, we use camera projection matrices to explicitly link camera- and object-centric coordinate frames. This allows us to reason about geometry and learn object priors in a common 3D coordinate system. More specifically, we use regressed camera pose information to back-project 2D feature maps to 3D feature grids at several scales. This is achieved within our novel architecture that comprises a 2D image encoder and a 3D shape decoder. They feature symmetrical downsampling and upsampling parts and communicate through multi-scale skip connections, as in the U-Net architecture [16]. However, unlike in other approaches, the bottleneck is made of 3D feature grids and we use back-projection layers [12, 11, 17] to lift 2D feature maps to 3D grids. As a result, feature localization from the input view is preserved. In other words, our feature embedding has a Euclidean structure and is aligned with object coordinate frame. Fig. 1 depicts this process. In reference to its characteristics, we dub our architecture UCLID-Net.

Earlier attempts at passing 2D features to a shape decoder via local feature extraction [24, 26] enabled spatial information to flow to the decoder in a non semantic manner, often with limited impact on the final result. In these approaches, the same local feature is attributed to all points lying along a camera ray. By contrast, UCLID-Net uses 3D convolutions to volumetrically process local features before passing them to the local shape decoders. This allows them to make different contributions at

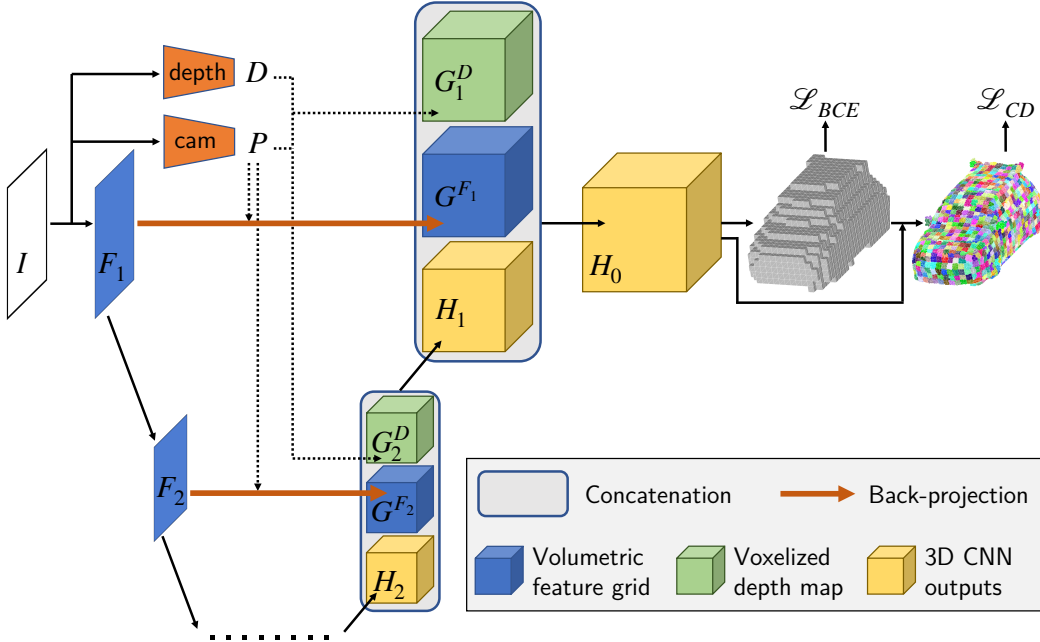

Figure 1: **UCLID-Net.** Given input image $I$, a CNN encoder estimates 2D feature maps $F_s$ for scales $s$ from 1 to $S$ while pre-trained CNNs regress a depth map $D$ and a camera pose $P$. $P$ is used to backproject the feature maps $F_s$ to object aligned 3D feature grids $G^{F_s}$ for $1 \le s \le S$ without using depth information. In parallel, $S$ corresponding voxelized depth grids $G_s^D$ are built from $D$ and $P$ without using feature information. A 3D CNN then aggregates feature and depth grids from the lowest to the highest resolution into outputs $H_S, \dots, H_0$ of increasing resolutions. From $H_0$, fully connected layers regress a coarse voxel shape, which is then refined into a point cloud using local patch foldings. Supervision comes in the form of binary cross-entropy on the coarse output and Chamfer distance on the final 3D point cloud.

different places along camera rays. To further promote geometrical reasoning, it never computes a global vector encoding of the input image. Instead, it relies on localized feature grids, either 2D in the image plane or 3D in object space. Finally, the geometric nature of the 3D feature grids enables us to exploit estimated depth maps and further boost reconstruction performance.

We demonstrate both on ShapeNet synthetic images, which are often used for benchmarking purposes, and on real-world images that our approach outperforms state-of-the-art ones. Our contribution is therefore a demonstration that creating a Euclidean preserving latent space provides a clear benefit for single-image reconstruction and a practical approach to taking advantage of it. Finally, the single-view pipeline naturally extends to multi-view reconstruction, which we also provide an example for.

## 2 Related work

Most recent SVR methods rely on a 2D-CNN to create an image description that is then passed to a 3D shape decoder that generates a 3D output. What differentiates them is the nature of their output which is strongly related to the structure of their shape decoder, and their approach to local feature extraction. We briefly describe these below.

**Shape Decoders** The first successful deep SVR models relied on 3D convolutions to regress voxelized shapes [5]. This restricts them to coarse resolutions because of their cubic computational and memory cost. This drawback can be mitigated using local subdivision schemes [9, 20]. MarrNet [25] and Pix3D [19] regress voxelized shapes as well but also incorporate depth, normal, and silhouette predictions as intermediate representations. They help disentangle shape from appearance and are used to compute a re-projection consistency loss. Depth, normal and silhouette are however not exploited in a geometric manner at inference time because they are encoded as flat vectors. PSGN [6] regresses sparse scalar values, directly interpreted as 3D coordinates of a point cloud with fixed

size and mild continuity. AtlasNet [8] introduces a per-patch surface parametrization and samples a point cloud from a set of learned parametric surfaces. One limitation, however, is that the patches it produces sometimes overlap each other or collapse during training [1].

To combine the strengths of voxel and mesh representations, Mesh R-CNN [7] uses a hybrid shape decoder that first regresses coarse voxels, which are then refined into mesh vertices using graph convolutions. Our approach is in the same spirit with two key differences. First, our coarse occupancy grid is used to instantiate folding patches and to sample 3D surface points in the AtlasNet [8] manner. However, unlike in AtlasNet, the locations of the sampled 3D points and the folding creating them are tightly coupled. Second, we regress shapes in object space, thus leveraging stronger object priors.

A competing approach is to rely on implicit shape representations. For example, the network of [14] computes occupancy maps that represent smooth watertight shapes at arbitrary resolutions. DISN [26] uses instead a Signed Distance Field (SDF). Shapes are encoded as zero-crossing of the field and explicit 3D meshes can be recovered using the Marching Cubes [13] algorithm.

**Local Feature Extraction** Most SVR methods discussed above rely on a vectorized embedding passing from image encoder to shape decoder. This embedding typically ignores image feature localization and produces a global image descriptor. As shown in [21], such approaches are therefore prone to behaving like classifiers that simply retrieve shapes from a learned catalog. Hence, no true geometric reasoning occurs and recognition occurs at the scale of whole objects while ignoring fine details.

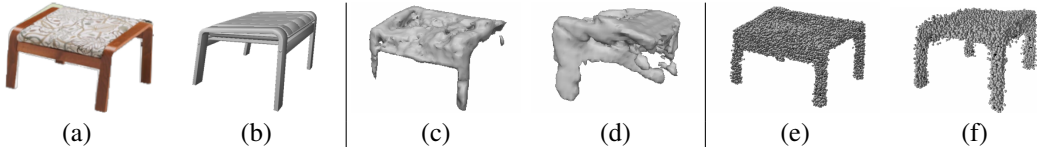

| (a) | (b) | (c) | (d) | (e) | (f) |

Figure 2: (a) Input photograph from Pix3D [19]. (b) Ground truth shape seen from a different viewpoint. (c,d) DISN [26] reconstruction seen from the viewpoints of (a) and (b), respectively. (e,f) Our reconstruction seen from the viewpoints of (a) and (b), respectively. For DISN, local feature extraction makes it easy to recover the silhouette in (c) but fails to deliver the required depth information. Our approach avoids this pitfall.

There have been several attempts at preserving feature localization from the input image by passing local vectors from 2D feature maps of the image encoder to the shape decoder. In [24, 7], features from the 2D plane are propagated to the mesh convolution network that operates in the camera space. In DISN [26], features from the 2D plane are extracted and serve as local inputs to a SDF regressor, directly in object space. Unfortunately, features extracted in this manner do not incorporate any notion of depth and local shape regressors get the same input all along a camera ray. As a result and as shown in Fig. 2, DISN can reconstruct shapes with the correct outline when projected in the original viewpoint but that are nevertheless incorrect. In practice, this occurs when the network relies on both global and local features, but not when it relies on global features only. In other words, it seems that local features allow the network to take an undesirable shortcut by making silhouette recovery excessively easy, especially when the background is uniform. The depth constraint is too weakly enforced by the latent space, and must be carried out by the fully connected network regressing signed distance value. By contrast, our approach does avoids this pitfall, as shown in Fig. 2(f). This is allowed by two key differences: (i) the shape decoder relies on 3D convolutions to handle global spatial arrangement before fully connected networks locally regress shape parts, and (ii) predicted depth maps are made available as inputs to the shape decoder.

## 3   Method

At the heart of UCLID-Net is a representation that preserves the Euclidean structure of the 3D world in which the shape we want to reconstruct lives. To encode the input image into it and then decode it into a 3D shape, we use the architecture depicted by Fig. 1. A CNN image encoder computes feature maps at $S$ different scales while auxiliary ones produce a depth map estimate $D$ and a camera projection model $P : \mathbb{R}^3 \rightarrow \mathbb{R}^2$. $P$ allows us to back-project image feature onto the 3D space along camera rays and $D$ to localize the features at the probable location of the surface on each of these ray.

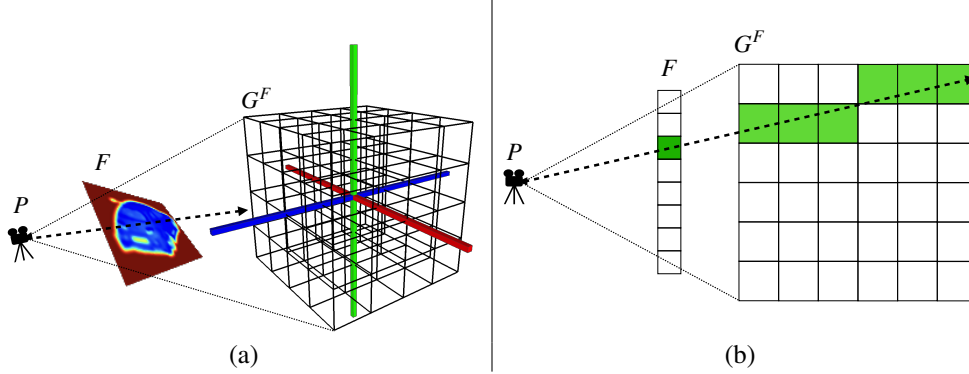

(a)                                                      (b)

Figure 3: (a) **Backprojecting 2D features maps to 3D grids.** Rays are cast from camera $P$ through 2D feature map $F$ to fill 3D grid $G^F$. It is applied to 2D feature maps from the image encoder to provide object space aligned 3D feature grids as inputs to the shape decoder (b) **Schematic view of a 1D to 2D backprojection:** all grid cells along a ray are given the same corresponding feature value.

The 2D feature maps and depth maps are back-projected to 3D grids that serve as input to the shape decoder, as shown by Fig. 3. This yields a coarse voxelized shape that is then refined into a point cloud. If estimates of either the pose $P$ or the depth map $D$ happen to be available *a priori*, we can use them instead of regressing them. We will show in the results section that this provides a small performance boost when they are accurate but not a very large one because our predictions tend to be good enough for our purposes, that is, lifting the features to the 3D grids.

The back-projection mechanism we use is depicted by Fig. 3. It is similar to the one of [12, 11, 17] and has a major weakness when used for single view reconstruction. All voxels along a camera ray receive the same feature information, which can result in failures such as the one depicted by Fig. 2 if passed as is to local shape decoders. To remedy this, we concatenate feature grids with voxelized depth maps. The result is then processed as a whole using 3D convolutions before being passed to local decoders. In the remainder of this section, we first introduce the basic back-projection mechanism, and then describe how our shape decoder fuses feature grids with depth information using a 3D CNN before locally regressing shapes.

### 3.1 Back-Projecting Feature and Depth Maps

We align all objects in the dataset to be canonically oriented within each class, centered at the origin, and scaled to fill bounding box $[-1, 1]^3$. Given such a 3D object, a CNN produces a 2D feature map $F \in \mathbb{R}^{f \times H \times W}$ for input image $I$. Using $P$, the camera projection used to render it into image $I \in \mathbb{R}^{3 \times H \times W}$, we back-project $F$ into object space as follows.

As in [12, 11], we subdivide bounding box $[-1, 1]^3$ into $G^F \in \mathbb{R}^{f \times N \times N \times N}$, a regular 3D grid. Each voxel $(x, y, z)$ contains the $f$-dimensional feature vector

$$G^F_{xyz} = F\{P \begin{pmatrix} x \\ y \\ z \end{pmatrix}\} , \tag{1}$$

where $\{\cdot\}$ denotes bilinear interpolation on the 2D feature map. As illustrated by Fig. 3, back-projecting can be understood as illuminating a grid of voxels with light rays that are cast by the camera and pass through the 2D feature map. This preserves geometric structure of the surface and 2D features are positioned consistently in 3D space.

In practice, we back-project 2D feature maps $(F_1, \ldots, F_S)$ of decreasing spatial resolutions, which yield 3D feature grids $(G^{F_1}, \ldots, G^{F_S})$ of decreasing sizes $(N_1, \ldots, N_S)$. We linearly scale the projected coordinates to account for decreasing resolution.

We process depth maps in a different manner to exploit the available depth value at each pixel. Given a 2D depth map $D \in \mathbb{R}_+^{H \times W}$ of an object seen from camera with projection matrix $P$, we first back-project the depth map to the corresponding 3D point cloud in object space. This point cloud is used to populate binary occupancy grids such as the one depicted by Fig. 4(a). As for feature maps,

we use this mechanism to produce a set of binary depth grids $(G_1^D, \ldots, G_S^D)$ of decreasing sizes $(N_1, \ldots, N_S)$.

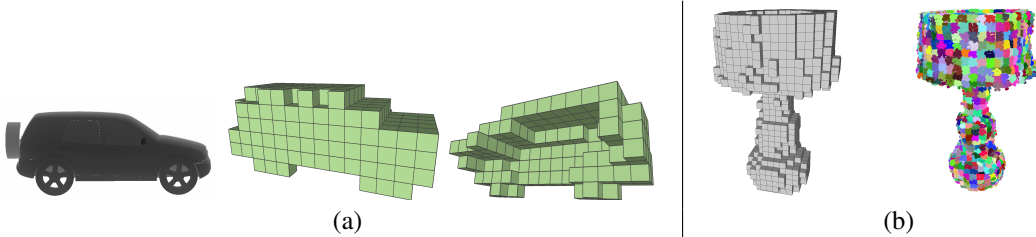

(a)                                                                              (b)

Figure 4: (a) **Back-projecting depth maps.** Input depth map and back-projected depth grid seen from two different view points. (b) **Outputs of the** $occ$ **and** $fold$ **MLPs** introduced in Section 3.2. One is an occupancy grid and the other a cloud of 3D points generated by individual folding patches. The points are colored according to which patch generated them.

## 3.2 Hybrid Shape Decoder

The feature grids discussed above contain learned features but lack an explicit notion of depth. The values in its voxels are the same along a camera ray. By contrast, the depth grids structurally carry depth information in a binary occupancy grid but without any explicit feature information. One approach to merging these two kinds of information would be to clamp projected features using depth. However, this is not optimal for two reasons. First, the depth maps can be imprecise and the decoder should learn to correct for that. Second, it can be advantageous to push feature information not only to the visible part of the surfaces but also to their occluded ones. Instead, we devised a shape decoder that takes as input the pairs of feature and depth grids at different scales $\{(G^{F_1}, G_1^D) \ldots, (G^{F_S}, G_S^D)\}$ we introduced in Section. 3.1 and outputs a point cloud.

Our decoder uses residual layers that rely on regular 3D convolutions and transposed ones to aggregate the input pairs in a bottom-up manner. We denote by $layer_s$ the layer at scale $s$, and $concat$ concatenation along the feature dimension of same size 3D grids. $layer_s$ takes as input a feature grid of size $N_s$ and outputs a grid $H_{s-1}$ of size $N_{s-1}$. If $N_{s-1} > N_s$, $layer_s$ performs upsampling, otherwise if $N_{s-1} = N_s$, the resolution remains unchanged. At the lowest scale, $layer_S$ constructs its output from feature grid $G^{F_S}$ and depth grid $G_S^D$ as

$$H_{S-1} = layer_S(concat(G^{F_S}, G_S^D)) . \tag{2}$$

At subsequent scales $1 \leq s < S$, the output of the previous layer is also used and we write

$$H_{s-1} = layer_s(concat(G^{F_s}, G_s^D, H_s)) . \tag{3}$$

The 3D convolutions ensure that voxels in the final feature grid $H_0$ can receive information emanating from different lines of sight and are therefore key to addressing the limitations of methods that only rely on local feature extraction [26]. $H_0$ is passed to two downstream Multi Layer Perceptrons (MLPs), we will refer to as $occ$ and $fold$. $occ$ returns a coarse surface occupancy grid. Within each voxel predicted to be occupied, $fold$ creates one local patch that refines the prediction of $occ$ and recovers high-frequency details in the manner of AtlasNet [8]. Both MLPs process each voxel of $H_0$ independently. Fig. 4(b) depicts their output in a specific case. We describe them in more detail in the supplementary material.

Let $\widetilde{O} = occ(H_0)$ be the occupancy grid generated by $occ$ and

$$\widetilde{X} = \bigcup_{\substack{xyz \\ \widetilde{O}_{xyz} > \tau}} \left\{ \begin{pmatrix} x \\ y \\ z \end{pmatrix} + fold(u, v | (H_0)_{xyz}) \mid (u, v) \in \Lambda \right\} \tag{4}$$

be the union of the point clouds generated by $fold$ in each individual $H_0$ voxel in which the occupancy is above a threshold $\tau$. As in [8, 27], $fold$ continuously maps a discrete set of 2D parameters $\Lambda \subset [0, 1]^2$ to 3D points in space, which makes it possible to sample it at any resolution. During the training, we minimize a weighted sum of the cross-entropy between $\widetilde{O}$ and the ground-truth surface occupancy and of the Chamfer-$L_2$ distance between $\widetilde{X}$ and a point cloud sampling of the ground-truth 3D model.

### 3.3 Implementation Details

In practice, our UCLID-Net architecture has $S = 4$ scales with grid sizes $N_1 = N_2 = 28$, $N_3 = 14$, $N_4 = 7$. The image encoder is a ResNet18 [10], in which we replaced the batch normalization layers by instance normalization ones [22]. Feature map $F_s$ is the output of the $s$-th residual layer. The shape decoder mirrors the encoder, but in the 3D domain. It uses residual blocks, with transposed convolutions to increase resolution when required. Last feature grid $H_0$ of the decoder has spatial resolution $N_0 = 28$, with 40 feature channels. The 8 first features serve as input to $occ$, and the last 32 to $fold$. $occ$ is made of a single fully connected layer while $fold$ comprises 7 and performs two successive folds as in [28]. The network is implemented in Pytorch, and trained for 150 epochs using the Adam optimizer, with initial learning rate $10^{-3}$, decreased to $10^{-4}$ after 100 epochs.

We take the camera to be a simple pinhole one with fixed intrinsic parameters and train a CNN to regress rotation and translation from RGB images. Its architecture and training are similar to what is described in [26] except we replaced its VGG-16 backbone by a ResNet18. To regress depth maps from images, we train another off-the-shelf CNN with a feature pyramid architecture [3]. These auxiliary networks are trained independently from the main UCLID-Net, but using the same training samples. Code is available here: `https://github.com/cvlab-epfl/UCLID-Net`.

## 4 Experiments

### 4.1 Experimental Setup

**Datasets.**   Given the difficulty of annotation, there are relatively few 3D datasets for geometric deep learning. We use the following two:

**ShapeNet Core** [2] features 38000 shapes belonging 13 object categories. Within each category objects are aligned with each other and we rescale them to fit into a $[-1, 1]^3$ bounding box. For training and validation purposes, we use the RGB renderings from 36 viewpoints provided in DISN [26] with more variation and higher resolution than those of [5]. We use the same testing and training splits but re-generated the depth maps because the provided ones are clipped along the z-axis.

**PIX3D** [19] is a collection of pairs of real images of furniture with ground truth 3D models and pose annotations. With 395 3D shapes and 10,069 images, it contains far less samples than ShapeNet. We therefore use it for validation only, on approximately 2.5k images of chairs.

**Baselines and Metrics.**   We test our UCLID-Net against several state-of-the-art approaches: AtlasNet [8] provides a set of 25 patches sampled as a point cloud, Pixel2Mesh [24] regresses a mesh with fixed topology, Mesh R-CNN [7] a mesh with varying topological structure, and DISN [26] uses an implicit shape representation in the form of a signed distance function. For Pixel2Mesh, we use the improved reimplementation from [7] with a deeper backbone, which we refer to as Pixel2Mesh+. All methods are retrained on the dataset described above, each according to their original training procedures.

We report our results and those of the baselines in terms of five separate metrics, Chamfer L1 and L2 Distances (CD–$L_1$, CD–$L_2$), Earth Mover's Distance (EMD), shell-IoU (sIoU), and average F-Score for a distance threshold of 5% (F@5%), which we describe in more detail in the supplementary material.

### 4.2 Comparative Results

**ShapeNet.**   In Fig. 5, we provide qualitative UCLID-Net reconstruction results. In Tab. 6(a), we compare it quantitatively against our baselines. UCLID-Net outperforms all other methods. We provide the results in aggregate and refer the interested reader to the supplementary material for per-category results. As in [26], all metrics are computed on shapes scaled to fit a unit radius sphere, and CD–$L_2$ and EMD values are scaled by $10^3$ and $10^2$, respectively. Note that these results were obtained using the depth maps and camera poses regressed by our auxiliary regressors. In other words, the input was only the image. We will see in the ablation study below that they can be further improved by supplying the ground-truth depth maps, which points towards a potential for further performance gains by using a more sophisticated depth regressor than the one we currently use.

**Pix3D.**   In Fig. 7, we provide qualitative UCLID-Net reconstruction results. In Tab. 6(b), we compare it quantitatively against our baselines. We conform to the evaluation protocol of [19] and

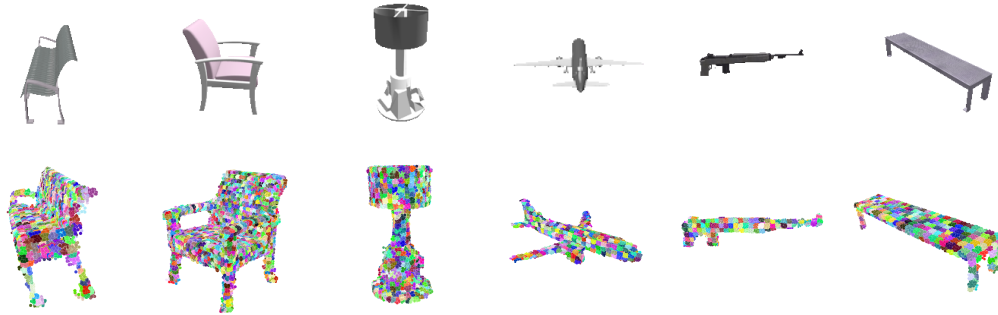

Figure 5: **ShapeNet objects reconstructed by UCLID-Net.** Top row: Input view. Bottom row: Final point cloud. The points are colored according to the patch that generated them.

| Method | CD-$L_2$ (↓) | EMD (↓) | sIoU (↑) | F@5% (↑) | Method | CD-$L_1$ (↓) | EMD (↓) |
|---|---|---|---|---|---|---|---|
| AtlasNet | 13.0 | 8.0 | 15 | 89.3 | Pix3D | 11.9 | 11.8 |
| Pixel2Mesh⁺ | 7.0 | 3.8 | 30 | 95.0 | AtlasNet | 12.5 | 12.8 |
| Mesh R-CNN | 9.0 | 4.7 | 24 | 92.5 | Pixel2Mesh⁺ | 10.0 | 12.3 |
| DISN | 9.7 | 2.6 | 30 | 90.7 | Mesh R-CNN | 10.8 | 13.7 |
| Ours | **6.3** | **2.5** | **37** | **96.2** | DISN | 10.4 | 11.7 |
| | | | | | Ours | **7.5** | **8.7** |

(ShapeNet)  (Pix3D)

Figure 6: **Comparative results.** For ShapeNet, we re-train and re-evaluate all methods. For Pix3D, lines 1-2 are duplicated from [19], while lines 3-6 depict our own evaluation using the same protocol. The up and down arrows next to the metric indicate whether a higher or lower value is better.

report the Chamfer-L1 distance (CD–$L_1$) and EMD on point clouds of size 1024. The CD–$L_1$ and EMD values are scaled by $10^2$. UCLID-Net again outperforms all other methods. The only difference with the ShapeNet case is that both DISN and UCLID-Net used the available camera models whereas none of the other methods leverages camera information.

## 4.3 From Single- to Multi-View Reconstruction

A further strength of UCLID-Net is that its internal feature representations make it suitable for multi-view reconstruction. Given depth and feature grids provided by the image encoder from multiple views of the same object, their simple point-wise addition at each scale enables us to combine them in a spatially relevant manner. For input views $a$ and $b$, the encoder produces feature/depth grids collections $\{(G_a^{F_1}, G_{1,a}^D) ..., (G_a^{F_S}, G_{S,a}^D)\}$ and $\{(G_b^{F_1}, G_{1,b}^D) ..., (G_b^{F_S}, G_{S,b}^D)\}$. In this setting, we feed $\{(G_a^{F_1} + G_b^{F_1}, G_{1,a}^D + G_{1,b}^D) ..., (G_a^{F_S} + G_b^{F_S}, G_{S,a}^D + G_{S,b}^D)\}$ to the shape decoder and let it merge details from both views. For best results, the decoder is fine-tuned to account for the change in magnitude of its inputs. As can be seen in Fig. 8, this delivers better reconstructions than those obtained from each view independently.

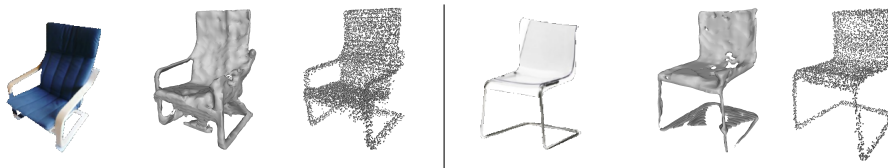

Figure 7: **Reconstructions on Pix3D photographs:** from left to right, twice: input, DISN, ours.

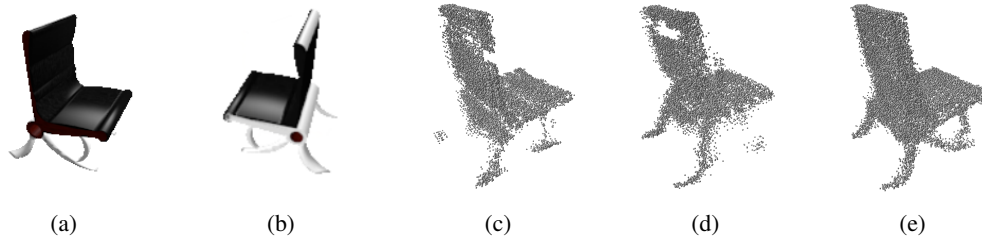

|   (a)   |   (b)   |   (c)   |   (d)   |   (e)   |

Figure 8: **Two-views reconstruction.** (a,b) Two input images of the same chair from ShapeNet. (c) Reconstruction using only the first one. (d) Reconstruction using only the second one. (e) Improved reconstruction using both images.

| Method: | depth | camera | CD-$L_2$($\downarrow$) | EMD($\downarrow$) |
|---------|-------|--------|-------------|---------|
| *CAR* | *inf.* | *inf.* | 4.08 | 2.23 |
| *NOD* | - | *GT* | 3.97 | 2.20 |
| *CAM* | *inf.* | *GT* | 3.83 | 2.16 |
| *CAD* | *GT* | *GT* | 3.80 | 2.14 |
| *ALL* | *inf.* | *inf.* | 4.03 | 2.23 |

|      (a)      |      (b)      |
|--------------|--------------|

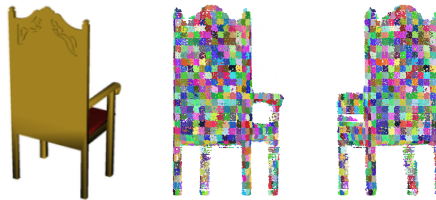

Figure 9: (a) **Ablation study:** comparative results on a single object category (cars) with inferred (*inf.*), ground truth (*GT*) or removed (-) auxiliary information. (b) **Failure mode.** From left-to-right: input view, reconstruction seen from the back-right, seen from the back-left. The visible armrest is correctly carved. The other one (occluded in the input) is mistakenly reconstructed as solid.

## 4.4 Ablation Study

To quantify the impact of regressing camera poses and depth maps, we conducted an ablation study on the ShapeNet car category. In Fig. 9(a), we report CD-$L_2$ and EMD for different network configurations. Here, *CAR* is trained and evaluated on the cars subset, with inferred depth maps and camera poses. *NOD* is trained and evaluated with ground truth camera poses, but without depth information (not ground truth nor regressed, we simply remove the depth branch). *CAM* is trained and evaluated with inferred depth maps, but ground truth camera poses. *CAD* is trained and evaluated with ground truth camera poses and depth maps. Finally, *ALL* is trained on 13 object categories with inferred depth maps and cameras as it was in all the experiments above, but evaluated on cars only.

Using ground truth data annotation for depth and pose improves reconstruction quality. The margin is not significant, which indicates that the regressed poses and depth maps are mostly good enough. Nevertheless, our pipeline is versatile enough to take advantage of additional information, such as depth map from a laser scanner or an accurate camera model obtained using classic photogrammetry techniques, when it is available. Fully removing the depth branch degrades accuracy. Note also that *ALL* marginally gets better performance than *CAR*. Training the network on multiple classes does not degrade performance when evaluated on a single class. In fact, having other categories in the training set increases the overall data volume, which seems to be beneficial.

In Fig. 9(b), we present an interesting failure case. The visible armrest is correctly carved out while the occluded one is reconstructed as being solid. While incorrect, this result indicates that UCLID-Net has the ability to reason locally and does not simply retrieve a shape from the training database, as described in [21].

## 5 Conclusion

We have shown that building intermediate representations that preserve the Euclidean structure of the 3D objects we try to model is beneficial. It enables us to outperform state-of-the-art approaches to single view reconstruction. We have also investigated the use of multiple-views for which our representations are also well suited. In future work, we will extend our approach to handle video sequences for which camera poses can be regressed using either SLAM-type methods or learning-based ones. We expect that the benefits we have observed in the single-view case will carry over and allow full scene reconstruction.

## Broader impact

Our work is relevant to a variety of applications. In robotics, autonomous camera-equipped agents, for which a volumetric estimate of the environment can be useful, would benefit from this. In medical applications, it would allow aggregating 2D scans to form 3D models of organs. It could also prove useful in industrial applications, such as in creating 3D designs from 2D sketches. More generally, constructing an easily handled differentiable representations of surfaces such as the ones we propose opens the way to assisted design and shape optimization.

As for any method enabling information extraction from images in an automated manner, malicious use is possible, especially raising privacy concerns. Accidents or malevolent use of autonomous agents is also a risk. To reduce accident threats we encourage the research community to propose explainable models, that perform more reconstruction than recognition - the latter regime arguably being more prone to adversarial attacks.

## Acknowledgment

This project was supported in part by the Swiss National Science Foundation.

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
