[Supplementary Material]

# Supplementary for: UCLID-Net: Single View Reconstruction in Object Space

## 1 Metrics

This section defines the metrics and loss functions used in the main paper.

### 1.1 Chamfer-L1

The Chamfer-L1 (CD–$L_1$) pseudo distance $d_{CD_1}$ between point clouds $X = \{x_i | 1 \leq i \leq N, x_i \in \mathbb{R}^3\}$ and $Y = \{y_j | 1 \leq j \leq M, y_j \in \mathbb{R}^3\}$ is the following:

$$d_{CD_1}(X,Y) = \frac{1}{|X|} \cdot \sum_{x \in X} \min_{y \in Y} \|x - y\|_2 + \frac{1}{|Y|} \cdot \sum_{y \in Y} \min_{x \in X} \|x - y\|_2, \qquad (1)$$

where $\|.\|_2$ is the Euclidean distance. We use CD–$L_1$ as a validation metric on the Pix3D dataset, according to the original procedure. It is applied on shapes normalized to bounding box $[-0.5, 0.5]^3$, and sampled with 1024 points.

### 1.2 Chamfer-L2

The Chamfer-L2 (CD–$L_2$) pseudo distance $d_{CD_2}$ between point clouds $X$ and $Y$ is the following:

$$d_{CD_2}(X,Y) = \frac{1}{|X|} \cdot \sum_{x \in X} \min_{y \in Y} \|x - y\|_2^2 + \frac{1}{|Y|} \cdot \sum_{y \in Y} \min_{x \in X} \|x - y\|_2^2 \qquad (2)$$

i.e. CD–$L_2$ is the average of the *squares* of closest neighbors matching distances. We use CD–$L_2$ as a validation metric on the ShapeNet dataset. It is applied on shapes normalized to unit radius sphere, and sampled with 2048 points.

### 1.3 Earth Mover's distance

The Earth Mover's Distance (EMD) is a distance that can be used to compare point clouds as well:

$$d_{EMD}(X,Y) = \min_{T \in \wp(N,M)} \sum_{1 \leq i \leq N, 1 \leq j \leq M} T_{i,j} \times \|x_i - y_j\|_2 \qquad (3)$$

where $\wp(N,M)$ is the set of all possible uniform *transport plans* from a point cloud of $N$ points to one of $M$ points, i.e. $\wp(N,M)$ is the set of all $N \times M$ matrices with real coefficients larger than or equal to 0, such that the sum of each line equals $1/N$ and the sum of each column equals $1/M$.

The high computational cost of EMD implies that it is mostly used for validation only, and in an approximated form. On ShapeNet, we use the implementation from [5] on point clouds normalized

to unit radius sphere, and sampled with 2048 points. On Pix3D, we use the implementation from [6] on point clouds normalized to bounding box $[-0.5, 0.5]^3$, and sampled with 1024 points.

## 1.4 F-score

The F-Score is introduced in [7], as an evaluation of distance between two object surfaces sampled as point clouds. Given a ground truth and a reconstructed surface, the F-Score at a given threshold distance $d$ is the harmonic mean of precision and recall, with:

- **precision** being the percentage of reconstructed points lying within distance $d$ to a point of the ground truth;
- **recall** being the percentage of ground truth points lying within distance $d$ to a point of the reconstructed surface.

We use the F-Score as a validation metric on the ShapeNet dataset. It is applied on shapes normalized to unit radius sphere, and sampled with 10000 points. The distance threshold is fixed at 5% side-length of bounding box $[-1, 1]^3$, i.e. $d = 0.1$ .

## 1.5 Shell Intersection over Union

We introduce shell-Intersection over Union (sIoU). It is the intersection over union computed on voxelized surfaces, obtained as the binary occupancy grids of reconstructed and ground truth shapes. As opposed to volumetric-IoU which is dominated by the interior parts of the objects, sIoU accounts only for the overlap between object surfaces instead of volumes.

We use the sIoU as a validation metric on the ShapeNet dataset. The occupancy grid divides the $[-1, 1]^3$ bounding box at resolution $50 \times 50 \times 50$, and is populated by shapes normalized to unit radius sphere.

# 2 Network details

We here present some details of the architecture and training procedure for UCLID-Net. We will make our entire code base publicly available.

**3D CNN** UCLID-Net uses $S = 4$ scales, and feature map $F_s$ is the output of the $s$-th residual layer of the ResNet18 [4] encoder, passed through a 2D convolution with kernel size 1 to reduce its feature channel dimension before being back-projected. In the 3D CNN, $layer_4$, $layer_3$, and $layer_2$ are composed of 3D convolutional blocks, mirroring the composition of a residual layer in the ResNet18 image encoder, with:

- 2D convolutions replaced by 3D convolutions;
- 2D downsampling layers replaced by 3D transposed convolutions.

Final $layer_1$ is a single 3D convolution. Each $concat$ operation repeats depth grids twice along their single binary feature dimension before concatenating them to feature grids. Tab. 1 summarizes the size of feature maps and grids appearing on Fig. 1 of the main paper.

**Local shape regressors** The last feature grid $H_0$ produced byt the 3D CNN is passed to two downstream Multi Layer Perceptrons (MLPs). First, a coarse voxel shape is predicted by MLP $occ$. Then, within each predicted occupied voxel, a local patch is folded in the manner of AtlasNet [3], by MLP $fold$. Both MLPs locally process each voxel of $H_0$ independently.

First, MLP $occ$ outputs a surface occupancy grid $\widetilde{O}$ such that

$$\widetilde{O}_{xyz} = occ((H_0)_{xyz}) \tag{4}$$

at every voxel location $(x, y, z)$. $\widetilde{O}$ is compared against ground truth occupancy grid $O$ using binary cross-entropy:

$$\mathcal{L}_{BCE}(\widetilde{O}, O) = -\sum_{xyz} \left[ O_{xyz} \cdot log(\widetilde{O}_{xyz}) + (1 - O_{xyz}) \cdot log(1 - \widetilde{O}_{xyz}) \right] \tag{5}$$

Table 1: **UCLID-Net architecture:** tensor sizes, names according to Fig. 1 of the main paper.

| Nature | Name | Spatial resolution | Number of features |
|---|---|---|---|
| input image | $I$ | $224 \times 224$ | 3 |
| 2D feature maps | $F_1$ | $56 \times 56$ | 30 |
| | $F_2$ | $28 \times 28$ | 30 |
| | $F_3$ | $14 \times 14$ | 30 |
| | $F_4$ | $7 \times 7$ | 290 |
| 2D feature grids | $G^{F_1}$ | $28 \times 28 \times 28$ | 30 |
| | $G^{F_2}$ | $28 \times 28 \times 28$ | 30 |
| | $G^{F_3}$ | $14 \times 14 \times 14$ | 30 |
| | $G^{F_4}$ | $7 \times 7 \times 7$ | 290 |
| 3D depth grids | $G_1^D$ | $28 \times 28 \times 28$ | |
| | $G_2^D$ | $28 \times 28 \times 28$ | 1 (binary) |
| | $G_3^D$ | $14 \times 14 \times 14$ | |
| | $G_4^D$ | $7 \times 7 \times 7$ | |
| 3D CNN outputs | $H_0$ | $28 \times 28 \times 28$ | 40 |
| | $H_1$ | $28 \times 28 \times 28$ | 73 |
| | $H_2$ | $28 \times 28 \times 28$ | 73 |
| | $H_3$ | $14 \times 14 \times 14$ | 146 |

$\mathcal{L}_{BCE}$ provides supervision for training the 2D image encoder convolutions, the 3D decoder convolutions and MLP *occ*.

Then $fold$, the second MLP learns a 2D parametrization of 3D surfaces within voxels whose predicted occupancy is larger than a threshold $\tau$. As in [3, 10], such learned parametrization is physically explained by folding a flat sheet of paper (or a patch) in space. It continuously maps a discrete set of 2D parameters $(u, v) \in \Lambda$ to 3D points in space. A patch can be sampled at arbitrary resolution. In our case, we use a single MLP whose input is locally conditioned on the value of $(H_0)_{xyz}$. The predicted point cloud $\widetilde{X}$ is defined as the union of all point samples over all folded patches:

$$\widetilde{X} = \bigcup_{\substack{xyz \\ \widetilde{O}_{xyz} > \tau}} \left\{ \begin{pmatrix} x \\ y \\ z \end{pmatrix} + fold(u, v | (H_0)_{xyz}) \mid (u, v) \in \Lambda \right\} \tag{6}$$

Notice that 3D points are expressed relatively to the coordinate of their voxel. As a result, we can explicitly restrict the spatial extent of a patch to the voxel it belongs to. We use the Chamfer-L2 pseudo-distance to compare $\widetilde{X}$ to a ground truth point cloud sampling of the shape $X$: $\mathcal{L}_{CD}(\widetilde{X}, X) = d_{CD_2}(\widetilde{X}, X)$.

$\mathcal{L}_{CD}$ provides supervision for training the 2D image encoder convolutions, the 3D decoder convolutions and MLP $fold$. The total loss function is a weighted combination of the two losses $\mathcal{L}_{BCE}$ and $\mathcal{L}_{CD}$. Practically, for training each patch of $\widetilde{X}$ is sampled with $|\Lambda| = 10$ uniformly sampled parameters, and $X$ is composed of 5000 points.

**Pre-training**   UCLID-Net is first trained for one epoch using the occupancy loss $\mathcal{L}_{BCE}$ only.

**Normalization layers**   In the ResNet18 that serves as our image encoder, we replace the batch-normalization layers by instance normalization ones. We empirically found out this provides greater stability during training, and improves final performance.

**Regressing depth maps**   We slightly adapt the off-the-shelf network architecture used for regressing depth maps [1]. We modify the backbone CNN to be a ResNet18 with instance normalization layers. Additionally, we perform less down-sampling by removing the initial pooling layer. As a result the input size is $224 \times 224$ and the output size is $112 \times 112$.

86 **Regressing cameras** We similarly adapt the off-the-shelf network architecture used for regressing
87 cameras in [9]: the backbone VGG is replaced by a ResNet18 with instance normalization layers.

## 3 Per-category results on ShapeNet

89 We here report per-category validation metrics for UCLID-Net and baseline methods: AtlasNet [3]
90 (AN), Pixel2Mesh⁺ and Mesh R-CNN [8, 2] (P2M⁺ and MRC), DISN [9] and UCLID-Net (ours).

91 Tab. 2 reports Chamfer-L2 validation metric, Tab. 3 the Earth Mover's Distance, Tab. 4 the Shell
92 Intersection over Union and Tab. 5 the F-Score at 5% distance threshold (ie. $d = 0.1$).

Table 2: **Chamfer-L2 Distance** (CD, $\times 10^3$) for single view reconstructions on ShapeNet Core, with various methods, computed on shapes scaled to fit unit radius sphere, sampled with 2048 points. The lower the better.

| method | plane | bench | box | car | chair | display | lamp | speaker | rifle | sofa | table | phone | boat | mean |
|---|---|---|---|---|---|---|---|---|---|---|---|---|---|---|
| AN | 10.6 | 15.0 | 30.7 | 10.0 | 11.6 | 17.3 | 17.0 | 22.0 | 6.4 | 11.9 | 12.3 | 12.2 | 10.7 | 13.0 |
| P2M⁺ | 11.0 | 4.6 | **6.8** | 5.3 | 6.1 | 8.0 | 11.4 | **10.3** | **4.3** | 6.5 | **6.3** | **5.0** | 7.2 | 7.0 |
| MRC | 12.1 | 7.5 | 9.7 | 6.5 | 8.9 | 9.3 | 14.0 | 13.5 | 5.7 | 7.7 | 8.1 | 6.9 | 8.6 | 9.0 |
| DISN | 6.3 | 6.6 | 11.3 | 5.3 | 9.6 | 8.6 | 23.6 | 14.5 | 4.4 | 6.0 | 12.5 | 5.2 | 7.8 | 9.7 |
| Ours | **5.3** | **4.2** | 7.4 | **4.1** | **4.7** | **6.9** | **10.9** | 13.8 | 5.8 | **5.7** | 6.9 | 6.0 | **5.0** | **6.3** |

Table 3: **Earth Mover's Distance** (EMD, $\times 10^2$) for single view reconstructions on ShapeNet Core, with various methods, computed on shapes scaled to fit unit radius sphere, sampled with 2048 points. The lower the better.

| method | plane | bench | box | car | chair | display | lamp | speaker | rifle | sofa | table | phone | boat | mean |
|---|---|---|---|---|---|---|---|---|---|---|---|---|---|---|
| AN | 6.3 | 7.9 | 9.5 | 8.3 | 7.8 | 8.8 | 9.8 | 10.2 | 6.6 | 8.2 | 7.8 | 9.9 | 7.1 | 8.0 |
| P2M⁺ | 4.4 | 3.2 | 3.4 | 3.4 | 3.7 | 3.7 | 5.5 | 4.2 | 3.5 | 3.4 | 3.8 | 2.7 | 3.4 | 3.8 |
| MRC | 5.0 | 4.1 | 5.1 | 4.1 | 4.7 | 4.9 | 5.6 | 5.7 | 4.1 | 4.6 | 4.5 | 4.6 | 4.2 | 4.7 |
| DISN | **2.2** | 2.3 | 3.2 | 2.4 | 2.8 | **2.5** | 3.9 | **3.1** | **1.9** | 2.3 | 2.9 | **1.9** | 2.3 | 2.6 |
| Ours | 2.5 | **2.2** | **3.0** | **2.2** | **2.3** | **2.5** | **3.2** | 3.4 | 2.0 | 2.4 | **2.7** | 2.2 | **2.2** | **2.5** |

Table 4: **Shell-Intersection over Union** (IoU, %) for single view reconstructions on ShapeNet Core, with various methods, computed on voxelized surfaces scaled to fit unit radius sphere. The higher the better.

| method | plane | bench | box | car | chair | display | lamp | speaker | rifle | sofa | table | phone | boat | mean |
|---|---|---|---|---|---|---|---|---|---|---|---|---|---|---|
| AN | 20 | 13 | 7 | 16 | 13 | 12 | 14 | 8 | 28 | 11 | 15 | 14 | 17 | 15 |
| P2M⁺ | 31 | 34 | 23 | 26 | 28 | 28 | 28 | 20 | 42 | 24 | 33 | 35 | 34 | 30 |
| MRC | 24 | 26 | 18 | 22 | 21 | 23 | 21 | 16 | 33 | 19 | 27 | 28 | 27 | 24 |
| DISN | 40 | 33 | 20 | 31 | 25 | **33** | 21 | 19 | **60** | 29 | 25 | **44** | 34 | 30 |
| Ours | **41** | **41** | **29** | **34** | **36** | **33** | **37** | **24** | 51 | **31** | **38** | 43 | **37** | **37** |

Table 5: **F-Score** (%) at threshold $d = 0.1$ for single view reconstructions on ShapeNet Core, with various methods, computed on shapes scaled to fit unit radius sphere, sampled with 10000 points. The higher the better.

| method | category | | | | | | | | | | | | | mean |
| --- | --- | --- | --- | --- | --- | --- | --- | --- | --- | --- | --- | --- | --- | --- |
| | plane | bench | box | car | chair | display | lamp | speaker | rifle | sofa | table | phone | boat | |
| AN | 91.2 | 85.9 | 73.8 | 94.4 | 90.5 | 84.3 | 81.4 | 79.7 | 95.6 | 91.1 | 90.8 | 90.4 | 90.3 | 89.3 |
| P2M⁺ | 90.3 | 97.1 | **96.0** | 97.9 | 95.7 | 93.1 | 90.2 | **91.3** | 96.8 | 96.5 | **95.8** | **97.6** | 94.4 | 95.0 |
| MRC | 88.4 | 93.3 | 92.1 | 96.4 | 92.0 | 91.4 | 85.8 | 88.3 | 94.9 | 95.0 | 93.9 | 95.9 | 92.8 | 92.5 |
| DISN | 94.4 | 94.3 | 88.8 | 96.2 | 90.2 | 91.8 | 77.9 | 85.4 | 96.3 | 95.7 | 86.6 | 96.4 | 93.0 | 90.7 |
| Ours | **96.1** | **97.5** | 94.3 | **98.5** | **97.4** | **95.8** | **92.7** | 90.6 | **98.0** | **97.0** | 95.5 | 96.4 | **97.1** | **96.2** |