[Reviews · NeurIPS 2020]

Review 1

Summary and Contributions: This work describes a method for 3D reconstruction (in the form of a point cloud) from a single image. It consists of two stages: 1) pre-training of depth and pose predictor networks (supervised); 2) training of an occupancy map and point cloud predictor network (also supervised; this is the main contribution). The authors show results on ShapeNet (synthetic) and also on Pix3D (real images).

Strengths: The paper is well written, and the proposed method seems solid. The motivation -- that local processing in 2D/3D and depth information help prevent degenerate outputs that conform to the silhouette but are wrong otherwise -- is well laid out, although not of a groundbreaking nature. The experimental results are very encouraging, both due to the improvements over previous methods on ShapeNet, but also due to the convincing demonstration of transfer to real images (Pix3D). Finally, the qualitative examples show both the proposed method (and some failure cases), and highlight the motivation (by showing failure cases of previous methods).

Weaknesses: The main weaknesses are the choice of supervision, which makes this problem much easier than those considered by past works, and some lack of novelty due to the reliance solely on existing and well-studied computational blocks. First of all, this method requires a 2-stage training, where pose and depth networks are first trained with direct supervision. This essentially "solves" most of 3D reconstruction from a single image, since merely projecting the depth and using the camera pose already gives a well-aligned, correct point cloud (though with missing points in occluded regions). The problem at hand is then one of shape completion, which is significantly easier. This is in contrast to several previous methods, which jointly estimate the camera pose, and many don't require depth supervision. As it stands, it may be that this work demonstrates the value of additional supervision modes (and using them correctly), rather than the value of the method itself. Secondly, it is difficult to place this paper in terms of novelty because all the building blocks have been proposed before. The main novel part seems to be the concatenation of the binary depth maps, which are not differentiable (and so couldn't be used in end-to-end learning), and which is in itself a well-known general computational block (a scatter operation). Contrasted to methods such as Differentiable Ray Consistency, which are older yet propose similar differentiable blocks, it seems less of a contribution. U-Nets and cascading feature pyramids are commonplace in computer vision, and several of the references given use a 2D-encoder-3D-decoder pipeline. Although well put together, this means that the proposed method is not extremely novel. The paper claim that this approach is better than distributing the features among all the points along a camera ray (like [22,24]), however this claim is not validated experimentally. An ablation without a depth predictor (i.e. not concatenating them to the feature maps) would verify this. A related, also unverified claim is that not having a global embedding is also responsible for the performance gains. However, the paper does not discuss the receptive field size, and compare it to the image and feature map sizes in order to turn this into a compelling argument.

Correctness: As far as I can tell, the method seems correct.

Clarity: The paper is clear, although the method section was hard to follow at times; it would require some polishing to improve its clarity. All other parts of the text are well written. The "fold" operation needs to be improved, in order to make the paper self-contained -- the explanation given is rushed and not enough to understand it without consulting the references. Fig. 3 is not clear and although the underlying operation can be understood from the main text, the caption does not describe it accurately and the graphics do not suggest the operation of placing each feature into voxel elements along the rays.

Relation to Prior Work: The cited prior work seems to be comprehensive enough, although there may be a few recent references that I missed.

Reproducibility: Yes

Additional Feedback: --- Post-rebuttal update --- I'd like to thank the authors for addressing my questions. In light of this response, I would like to stand by my initial recommendation.


Review 2

Summary and Contributions: This paper tackles the task of single (or multi) view 3D inference. Given an input image, the proposed approach: a) predicts 2D feature, global camera pose and pixel wise depth, b) projects depth and pixel based features to a canonical 3D volume (at multiple scales), c) predicts a coarse to fine 3D feature which is subsequently decoded to a per-voxel occupancy, and d) predicts a local point-cloud for each occupied voxel. In comparison to prior work, which typically does not incorporate depth based reasoning or local point-cloud prediction, this approach is shown to yield stronger empirical results on ShapenNet and Pix3D datasets.

Strengths: - While previous approaches have looked at propagating 2D features to 3D voxels/points, they typically do not account for visibility i.e. all points/voxels along a ray get similar 2D features. By incorporating reasoning about depth, this approach allows overcoming this restriction. - The empirical results, in particular the improvements over the related approach of DISN which follows a similar high-level pipeline, demonstrate the benefits of depth based features, multi-scale reasoning and per-voxel point cloud generation. - The ablations demonstrating the effects of GT-vs-predicted depth/camera are interesting and seem to imply that the gains with GT over predicted are surprisingly small i.e. using predicted depth is good enough.

Weaknesses: - While my current rating gives the paper the benefits of the doubt, several aspects of the evaluation are unclear to me. I would want certain things clarified regarding why the evaluation reported here is significantly different from other papers e.g. Mesh RCNN. In particular: a) When comparing the CD evaluation reported in this work against the evaluation in mesh-RCNN for ShapeNet (Tables 1 and 2), the reported numbers differ by an order of magnitude! I think that work also reports CD-L2 so I am perplexed at the difference and would really appreciate an explanation (or even better, evaluation of this paper's predictions using the Mesh-RCNN benchmarking code). b) Typically, when computing CD (and EMD), points are only sampled on mesh surface. However, it seems this approach also generates points in interior (as occ=1 for the interior voxels, so Eqn 4 would also generate points for these)? Doesn't this hurt the CD evaluation as points in ground-truth would only be sampled on the mesh surface? Or is some different procedure followed to generate the predicted and GT point samples? - As mentioned above in strength 2, the overall approach is shown to be better than the relevant baseline e.g. DISN. However, there are several components to the approach and their importance is not disentangled. For example: a) Eqn 1 implies the feature for each pixel is copied to a single 3D location unlike previous approaches where a pixel's features are copied to all 3D locations along a ray. How important is this difference when creating G^F. b) In Eq 2 and 3, what if we do not use G^D (i.e. depth based features)? The text argues this is important but this is not empirically shown. c) Eqn 4 implies multiple points are generated per occupied voxel - is this crucial? Overall, while the paper proposes a system that work well, the experiments to do not disentangle which aspects of this system are important. - The proposed approach crucially relies on predicting depth and camera pose for relating pixel to their corresponding canonical coordinates. While this is possible to do in synthetically generated data with limited camera variation, this prediction is much harder (and ambiguous) depth prediction is ambiguous in real scenes e.g. for a far-away bird, how can we exactly predict per-pixel depth and camera?

Correctness: The technical details and empirical setup seem correct. I have some concerns regarding the evaluation details though (mentioned in Weakness 1).

Clarity: The paper is generally well written and easy to follow.

Relation to Prior Work: I think the relation to prior work is presented accurately.

Reproducibility: Yes

Additional Feedback: Overall, assuming the concerns regarding the evaluation will be addressed, this paper presents a useful system that makes several intuitive changes in comparison to prior work. While the paper would have been stronger had the individual effects of these been disentangled, I would still lean towards acceptance. ---- Thanks for the response - it does resolve my concerns regarding the evaluation metric and the ablations reported are also helpful. I would like to keep the current rating of leaning towards accept.


Review 3

Summary and Contributions: This work proposes a novel architecture for single view 3D shape reconstruction. The main novelties of the paper are: lifting the 2D features extracted from the image to a 3D space by using either a regressed or provided camera pose and depth map, and creating the final point cloud with 3D convolutions and an approach similar to AtlasNet/FoldingNet but applied to local neighborhoods only. The final output of the model is a point cloud composed of many patches stitched together according to an occupancy map. The method shows good results both on the synthetic ShapeNet dataset and on the realistic Pix3D dataset.

Strengths: + The idea of regressing a full point cloud as the stitching of local patches regressed from a 3D volume is quite interesting and seems to produce quite nice and detailed results. + The authors show how the same method can be easily extended to multi-view reconstruction without requiring a retraining of the whole system, but just of the decoder part. + The method seems to be able to generalize reasonably well to real objects outside of the training domain.

Weaknesses: - The method requires images acquired by a calibrated camera to backproject the 2D features to the 3D grid. Most competing methods do not have this assumption. - The subdivision of tasks between the creation of the occupancy map and the regression of the pointcloud in the decoder is quite interesting and seems effective. However it exposes the method to an obvious possible failure: if the occupancy map is not regressed correctly the point cloud part of the architecture has no way of fixing the prediction. To my understanding, in the case of an occupied voxel being predicted as empty the local folded patch of point will be completely ignored, while for an empty voxel predicted as full the predicted point will be inevitably considered and will add unwanted points to the final prediction. - The main advantage of the method seems to come from reasoning locally in the voxel grid space and creating the final point cloud as the stitch of many local patches (way more than the main competitors were able to do before). The focus on local regions however can also be one of the drawbacks of the method since contiguity between predicted nearby patches is not enforced in any way. The method might easily generate shapes that are not properly “closed” at the intersection between regressed patches. A comment by the authors on this point will be interesting. - Some additional ablation studies would have been nice to motivate the design choices made by the authors. For example what happens experimentally if you clamp projected features using depth? The explanation between line 136 and 142 it’s reasonable but I would have liked to see it supported by experimental results. Moreover what is the impact of using multiple scales for the elaboration? How much would the performance degrade using a single scale? - On the real Pix3D dataset the method is tested only on the chair class, is there any reason to not test on all classes? - Some implementation details of the method should be made more clear, but I think that this weakness can be addressed in a revised version of the paper, see my questions to the authors below.

Correctness: I believe most claims of the paper to be correct and the experimental evaluation to be fair.

Clarity: The paper is well written and easy to follow, some implementation details are missing but they might be add in a revised version of the paper

Relation to Prior Work: Most previous works have been cited, however some of the implicit shape competing methods have not been included into the experimental evaluation, i.e. [14] and [4].

Reproducibility: Yes

Additional Feedback: I’m giving this work a 7 because I consider it a good submission, but I would like to receive some clarification from the authors which should consider adding them to the manuscript: (a.) All the 2D networks of the proposed method use a resnet18 backbone, i.e. the depth prediction, pose estimation and feature extraction networks. Is it a single shared backbone or each subnetwork has its own set of parameters? (b.) At the end of the architecture the 40 final features are splitted into 8 and 32 and then sent to the `occ` and `fold` branches of your architecture. Why this split? Wouldn’t have been simpler to use the full 40 features for both branches? (c.) How is the threshold \tau of the occupancy grid selected? ~~~ Post rebuttal comments ~~~ I have read all the reviews and the authors' rebuttal and I'm leaning towards keeping my original acceptance rating. My doubts have been properly addressed by the authors in the rebuttall and the evaluation concerns raised by R2 and the ablation experiments suggested by R1 seems to have been adressed as well. I do agree with R1 that most of the components are not novel, but the way they are combined seems novel to me and the results on the real Pix3D dataset are encouraging.

[Author Response · NeurIPS 2020]

We thank the reviewers for their insightful feedback. We address their concerns below.

**R1.Q1: Supervision Level.** Existing single view reconstruction methods we compare to use synthetic renderings of
3D meshes for training. Therefore, camera poses and depth maps are available for free in this experimental setup.
In effect, we propose to make use of this additional information to break down 3D reconstruction into simpler tasks.
Similarly, in a scenario where ground truth 3D models are used in association with real photographs, well-established
computer vision methods can be used to allow for recovering ground truth camera pose and depth map, as was done in
the pix3D paper. There is therefore not much benefit in *not* taking advantage of them.

**R1.Q2: Novelty.** We acknowledge that we do not introduce any entirely new computational block. The novelty lies in
how we assemble the blocks in a principled manner to break down 3D reconstruction into simpler subtasks. The hybrid
shape decoder is also novel and allows for coarse-to-fine reconstruction.

**R1.Q3: Depth Ablation Study.** All reviewers noted that an ablation study on
the influence of appending depth predictions to 3D feature grids was missing.
In Tab. 1, we therefore report reconstruction accuracy on the subset of cars in the setting
where ground truth camera poses are known, for different depth predictors: Using ground
truth depth maps (*GT*), inferring them (*INF*), and removing them from the pipeline (*NO*).
Removing depth information significantly degrades accuracy and using our inferred depth
maps delivers an accuracy approaching that using the ground truth ones.

| Depth | CD($\downarrow$) | EMD($\downarrow$) |
|---|---|---|
| *GT* | 3.80 | 2.14 |
| *INF* | 3.83 | 2.16 |
| *NO* | 3.97 | 2.20 |

Table 1: **Depth ablation**

**R1.Q4: Embedding.** Our intuition is that relying on a 1D flat vector embedding forces most 3D reconstruction
networks to semantically encode localization information, and the reviewer has a point that we do not demonstrate
this explicitly. This is not easy to do because using 1D global embeddings would require a totally different network
architecture. We would welcome any suggestion on how to do this properly.

**R1.Q5: Clarity.** We will clarify to make the paper self-contained, and improve Fig. 3.

**R2.Q1a: Benchmark metrics.** In Tab. 2 of Mesh-RCNN, shapes are normalized to bounding boxes of side length 10,
and sampled using 10k points. In contrast and as detailed in the supplementary, we normalize to unit sphere, sample
2048 points and scale the final CD-L2 by a factor of $10^3$ (like DISN). This strongly affects CD-L2 since it is dependent
on scale and sampling density. Using the Mesh-RCNN setting on our dataset, our method yields an average CD-L2 of
0.197 to be compared with 0.250 for Mesh-RCNN.

**R2.Q1b: Occupancy.** We only sample points on the object's surface, without any points being generated inside.
Therefore, in our occupancy maps, $occ = 1$ *only* for voxels intersecting the surface.

**R2.Q2a: Back-projection.** This is a misunderstanding. The features at a 2D location are back-projected everywhere
along the camera ray, without reference to depth. We will clarify.

**R2.Q2b: Depth Ablation.** See R1.Q3 and Tab. 1.

**R2.Q2c: Per-Voxel Point Sampling.** It is essential to sample multiple points per voxels, since otherwise output
shapes would be voxelized at a relatively coarse resolution ($28^3$). The effect of this refinement using folded patches is
qualitatively shown shown in Fig. 4b of the main paper. Quantitatively, on our whole testing set, sampling points at the
center of occupied voxels instead of locally folding patches yields an increase of ~6.3 in EMD.

**R2.Q3: Synthetic vs. Real Scenes** We share the reviewer's concern and would add that it applies to most current
single view reconstruction deep learning methods, many of which only work on clean synthetic images. This is why we
tested ours on the pix3D dataset with real images that feature more complex shadows, textures, and exposures. We will
focus on the other issues the reviewer mentions in future work.

**R3.Q1: Supervision Level.** See R1.Q1.

**R3.Q2: Failure to Predict Occupancy.** In case occupancy is wrongly predicted, there is indeed no way for the local
patch folders to recover the correct shape. For this reason, a strong emphasis is put on occupancy during training:
during the first epoch the network is supervised using $\mathcal{L}_{BCE}$ only. Then the total loss is $\mathcal{L} = 100 * \mathcal{L}_{BCE} + \mathcal{L}_{CD}$.

**R3.Q3: Ensuring Local Patches Contiguity.** Patches sometimes do not perfectly align at voxels' borders. We will fix
this in future work using either a regularizer, or different architecture, or downstream deterministic computations.

**R3.Q4: Ablation Study.** We provide additional ablation studies to support design choices: see R1.Q3 and Tab. 1 for
an ablation of depth information. In addition, hard clamping the 3D feature grids using depth maps incurs an increase
of ~0.16 in CD and ~0.07 in EMD compared to simply appending them to 3D feature grids for 3D convolutions.

**R3.Q5: Pix3D Benchmark.** We only tested on the chair subset of pix3D because it is the benchmark for single view
reconstruction proposed in the original pix3D paper. Our method generalizes to other classes as shown in Fig. 1. For
tables, we get a CD-L1 of 7.7 compared to 7.5 on chairs.

**R3.Q6: Implementation Details.** We will clarify the following points: **a)** Depth
prediction, pose estimation and feature extraction subnetworks each have their own
set of parameters. **b)** The 40 final features are split into 8 and 32 and then sent to
the *occ* and *fold* branches with the intent to encourage disentanglement between
occupancy and local patch deformation. Early experiments showed better results
over feeding the entire set of features. **c)** The occupancy threshold $\tau$ is manually
tuned at 0.35.

Figure 1: **Pix3D table: input image and 3D shape reconstructed by our method.**

[Meta-Review · NeurIPS 2020]

Three expert reviewers agree that the paper can be accepted to the conference. The authors' rebuttal did a good job at addressing the reviewers' concerns about the details of the experimental evaluation. I recommend acceptance.